# Management of biliary stricture in patients with IgG4-related sclerosing cholangitis

**Masaki Miyazawa**[1]*, **Hajime Takatori**[1], **Kazunori Kawaguchi**[1], **Kazuya Kitamura**[1],
**Kuniaki Arai**[1], **Koichiro Matsuda**[2‡], **Takeshi Urabe**[3‡], **Katsuhisa Inamura**[4‡],
**Takuya Komura**[5‡], **Hideki Mizuno**[6‡], **Uichiro Fuchizaki**[7‡], **Taro Yamashita**[1],
**Yoshio Sakai**[1], **Tatsuya Yamashita**[1], **Eishiro Mizukoshi**[1], **Masao Honda**[1],
**Shuichi Kaneko**[1]

1 Department of gastroenterology, Kanazawa University Hospital, Kanazawa, Japan, 2 Department of
internal medicine, Toyama Prefectural Central Hospital, Toyama, Japan, 3 Department of gastroenterology,
Public Central Hospital of Matto Ishikawa, Hakusan, Japan, 4 Department of internal medicine, Tonami
General Hospital, Tonami, Japan, 5 Department of gastroenterology, National Hospital Organization
Kanazawa Medical Center, Kanazawa, Japan, 6 Department of gastroenterology, Toyama City Hospital,
Toyama, Japan, 7 Department of gastroenterology, Keiju Medical Center, Nanao, Japan

☯ These authors contributed equally to this work.
‡ These authors also contributed equally to this work.
* miyacchi_1985@yahoo.co.jp

pone.0232089

UNITED STATES

**Data Availability Statement:** Data cannot be
shared publicly because the data that contains
sensitive patient information must not be made
publicly available. The data set, "IgG4-SC

## Abstract

### Background

We aimed to determine the optimal approach with endoscopic biliary drainage (EBD) and
corticosteroid (CS) for the treatment of IgG4-related sclerosing cholangitis (ISC).

### Methods

To evaluate the safety of EBD for treatment of biliary stricture caused by ISC, we assessed
the risk of stent dislodgement and sought to determine the most appropriate time for stent
removal. We also assessed the safety of treatment with CS alone for patients with obstruc-
tive jaundice, and the rate of and risk factors for biliary tract complications.

### Results

Sixty-nine patients with ISC treated with CS were enrolled. Twenty-eight patients (40.6%)
were treated with EBD for biliary stricture before CS initiation. Intentional stent removal was
performed in thirteen (46.4%) after confirming CS-induced improvement. Eleven of thirteen
patients (84.6%) underwent stent removal within 1 month after CS initiation and all their
stent removals were safely carried out without early (within two weeks) recurrence of
obstructive jaundice. Ten of twenty-eight patients (35.7%) experienced spontaneous stent
dislodgement after CS initiation, and seven (70%) of them developed stent dislodgement
two weeks to two months after CS initiation. Among forty-one patients treated with CS alone
without EBD, 10 patients had obstructive jaundice at the time of CS initiation and all of them
achieved clinical improvement without biliary tract infection. During the follow-up, three
patients (4.3%), all of whom had undergone EBD, developed bile-duct stones, while none of

Kanazawa," is available from Ethics Committee of Kanazawa University Hospital (approval number 2016-191) for researchers who meet the criteria for access to confidential data. Please contact via phone at +81-76-265-2233. The Ethics Committee of Kanazawa University Hospital can also be contacted via email at rinri@adm.kanazawa-u.ac.jp.

**Funding:** The authors received no specific funding for this work.

**Competing interests:** The authors have declared that no competing interests exist.

those treated with CS alone developed bile-duct stones ($p$ = 0.032). Long-term biliary stenting was a risk factor for bile-duct stones.

## Conclusions

Biliary stent removal should be carried out within 2 weeks after CS initiation if biliary stricture improves to prevent stent dislodgement. Obstructive jaundice can be treated safely with CS alone in patients without infection. Clinicians should be aware of the possibility of bile-duct stones in patients treated with EBD.

## Introduction

IgG4-related sclerosing cholangitis (ISC) is characterized by an elevated serum IgG4 level and IgG4-positive plasma cell infiltration of the bile-duct wall[1]. ISC is one of the many phenotypes of systemic IgG4-related diseases. ISC is frequently associated with autoimmune pancreatitis (AIP)[2], although some patients with ISC do not have involvement of other organs[3].

ISC frequently causes biliary stricture and obstructive jaundice as sequelae to fibro-inflammatory changes in the bile-duct wall. Similar to AIP, ISC displays a good response to corticosteroid (CS) therapy[4,5]. The clinical and radiological features of ISC are resolved by CS in most patients. However, CS may worsen obstructive jaundice and trigger biliary tract infection. Therefore, most patients with obstructive jaundice undergo endoscopic biliary drainage (EBD) before CS initiation[6,7].

EBD and CS play an important role in the treatment of biliary stricture in patients with ISC, and they influence the short- and long-term prognosis. The optimal timing of stent removal, to prevent spontaneous stent dislodgement due to improved biliary stricture or early recurrence of obstructive jaundice caused by residual biliary stricture, is unclear. Data regarding the safety and effectiveness of treatment with CS alone (*i.e.*, without EBD) for obstructive jaundice are insufficient. Also, the risk of biliary tract complications during long-term management of ISC is unknown. We aimed to determine the optimal approach for management of biliary stricture in patients with ISC.

## Methods

### Subjects

This retrospective study was approved by the Ethics Committee of Kanazawa University Hospital (the approval number is 2016–191). We enrolled patients diagnosed with ISC and treated with CS in our hospital or its affiliated institutes from January 2008 to December 2018. The diagnosis of ISC was based on the 2012 clinical criteria of the Japanese working group[8]. The enrolled patients satisfied the criteria for definite or probable ISC and were followed up for > 6 months. Patients who underwent surgical resection or received other immunosuppressive agents were excluded. The medical records of the patients were reviewed, and their clinical data were collected. We evaluated the following clinical parameters: age, sex, serum total bilirubin (TB) level, IgG and IgG4 level, endoscopic retrograde cholangiography (ERC) findings (type 1/2/3/4) based on ISC-specific diagnostic criteria[1], the incidence and type (diffuse/local) of AIP, treatment modality (EBD and/or CS), and the incidence of complications during the follow-up.

### Treatment and follow-up of ISC

The enrolled patients underwent ERC for diagnosis of biliary stricture, exclusive diagnosis of biliary tract cancer, and/or treatment with EBD. In this study, patients who received endoscopic nasobiliary drainage (ENBD) as initial treatment with EBD were excluded because the status of ENBD insertion must not have let us evaluate the incidence of spontaneous stent dislodgement and its precise time after CS initiation.

CS was started after ISC diagnosis in almost all patients, while some received CS as an optional adjunct treatment. CS therapy started with prednisolone at 20–40 mg/day, which was tapered by 2.5–10 mg every 1–4 weeks. We evaluated clinical improvement during tapering of the prednisolone dose by assessing the thickness of the bile-duct wall using computed tomography (CT) and/or according to the degree of biliary stricture by ERC within 1 month of initiating CS. A variety of CS regimens were used for maintenance therapy, based on the patient's condition; maintenance therapy was defined as continuous administration of CS for > 6 months after initiation. A maintenance dose was defined as a dose of daily prednisolone administered for the longest period during the follow-up.

After CS initiation in patients with ISC treated with EBD using a plastic stent for biliary stricture, many of them underwent planned stent removal after improvement of ISC; in other cases the stent was not removed. Some of the patients with prolonged biliary stenting experienced spontaneous stent dislodgement (confirmed by radiological imaging) during the follow-up. To evaluate the effectiveness and safety of EBD, we noted the time from CS initiation to planned stent removal or spontaneous stent dislodgement. We could not determine the precise time of spontaneous stent dislodgement, so it was defined as the halfway point between the last time a biliary stent was seen and the first time its dislodgement was confirmed.

We assessed the rate of clinical relapse of ISC, which was defined as reappearance of symptoms and/or repeated elevation of serum biliary enzyme levels in addition to exacerbation of biliary stricture on radiological imaging after CS-induced improvement. We also noted the occurrence of biliary tract complications, such as bile-duct stones, biliary tract cancer, and infectious diseases, during the follow-up.

### Statistical analysis

Categorical variables were compared by chi-squared test, and continuous variables by Student's *t*-test or the Mann–Whitney *U* test. Differences with *p*-values < 0.05 were considered statistically significant. Statistical analysis was performed using StatView software (SAS Institute, Cary, NC, USA).

## Results

### Characteristics of the patients with ISC

This study included 69 patients with ISC treated with CS. The median follow-up period was 65.3 months (range: 7.0–192.1 months). The clinical characteristics of the patients are listed in Table 1. Based on the 2012 clinical criteria[8], 68 patients (98.6%) were judged to have definitive ISC and 1 patient (1.4%) to have probable ISC. Thirty-two patients (46.4%) had obstructive jaundice with a serum TB level of > 3.0 mg/dL. The serum IgG level was elevated to > 1,800 mg/dL in 39 of 66 patients (59.1%), and the serum IgG4 level was elevated to > 135 mg/dL in all 69 of the patients (100%) examined. Regarding the cholangiographic type of ISC, type 1 was detected in 55 patients (79.7%), type 2 in 5 (7.2%), type 3 in 7 (10.1%), and type 4 in 2 patients (2.9%). That is, extrapancreatic biliary stricture was seen in 14 patients (20.3%). Forty-eight (69.6%) and twenty (29.0%) patients developed diffuse and focal AIP, respectively.

**Table 1. The clinical characteristics of patients with IgG4-related sclerosing cholangitis.**

| Parameter (*n* = 69) | |
|---|---|
| Age, mean ± SD, years | 66.1 ± 10.5 |
| Sex, male / female (%) | 60 (87.0) / 9 (13.0) |
| Accuracy of diagnosis, definite / probable | 68 (98.6) / 1 (1.4) |
| Obstructive jaundice[†], present / absent (%) | 32 (46.4) / 37 (53.6) |
| Serum IgG level, median (range), mg/dl | 1871.5 (950–5963) |
| Serum IgG4 level, median (range), mg/dl | 484 (145–4080) |
| Cholangiography findings, type 1 / 2 / 3 / 4 | 55 / 5 / 7 / 2 |
| Concurrence of autoimmune pancreatitis, diffuse / focal / absent | 48 / 20 / 1 |
| Endoscopic biliary drainage, present / absent (%) | 28 (40.6) / 41 (59.4) |
| CS maintenance therapy[‡], present / absent (%) | 64 (92.8) / 5 (7.2) |
| Maintenance dose of prednisolone, median (range), mg/day | 5 (1–10) |
| Follow-up period[§], median (range), months | 65.3 (7.0–192.1) |

† Serum total bilirubin level of > 3.0 mg/dL with dilation of bile-duct.

‡ Maintenance therapy was defined as continuous administration of CS for > 6 months after initiation.

§ Follow-up period counts from CS initiation.

CS, corticosteroid therapy.

## Treatment with EBD and/or CS

Twenty-eight patients (40.6%) were treated with EBD for biliary stricture before CS initiation (Table 1) and received placement of 7-French straight-type plastic stents. Of the patients treated with EBD, type 1 was seen in 23 patients, type 2 in 2, and type 3 in 3 patients classified by ERC findings. Of the 5 patients of type 2 or 3 ISC, who had multiple biliary stricture, 3 received placement of long plastic stents that reached the intrahepatic bile-duct because drainage for severe extrapancreatic biliary stricture that caused obstructive jaundice was necessary. No patient had symptoms of purulent cholangitis such as fever or chills before EBD.

The management of EBD is detailed in Table 2. Intentional stent removal was performed in 13 of 28 patients (46.4%) after confirming CS-induced improvement. The median time from CS initiation to stent removal was 16 days (range: 10–143 days). Eleven of thirteen patients (84.6%) underwent stent removal within 1 month of CS initiation. The median daily prednisolone dose at the time of stent removal was 30 mg (range: 10–40 mg). Stent removal was safely carried out without early (within 2 weeks) recurrence of obstructive jaundice in all 11 patients who underwent stent removal within 1 month of CS initiation. Fifteen of 28 patients (53.6%) underwent prolonged biliary stenting and twelve (42.9%) experienced spontaneous stent dislodgement. Two and ten of those twelve patients experienced spontaneous stent dislodgement before and after CS initiation, respectively. The estimated median time from CS initiation to

**Table 2. The management of biliary stricture for IgG4-related sclerosing cholangitis using endoscopic biliary drainage.**

| Stent removal / dislodgement | Time from CS initiation to stent removal / stent dislodgement, median (range), days | Daily prednisolone dose at the time of stent removal / dislodgement, median (range), mg/day |
|---|---|---|
| Removal (*n* = 13) | 16 (10–143) | 30 (10–40) |
| Dislodgement (*n* = 10) | 39 (9–154) [†] | 15 (5–30) |

† The time of spontaneous stent dislodgement was defined as the halfway point between the last time a biliary stent was seen and the first time its dropout was confirmed.

CS, corticosteroid therapy.

spontaneous stent dislodgement in the 10 patients was 39 days (range: 9–154 days), and the median daily prednisolone dose at the estimated time of spontaneous stent dislodgement was 15 mg (range: 5–30 mg). Seven of those ten patients (70%) developed spontaneous stent dislodgement 2 weeks to 2 months after CS initiation. None of the patients developed complications associated with stent dislodgement; *e.g.*, intestinal perforation.

All 69 patients were treated with CS for a median of 34.4 months (range: 1.4–160.7 months). Sixty-four patients (92.8%) received CS maintenance therapy for $> 6$ months. The median maintenance dose of oral prednisolone was 5 mg (range: 1–10 mg). Forty-one patients (59.4%) were treated with CS alone. The differences in clinical characteristics between the patients treated with CS alone and CS in addition to EBD are shown in Table 3. The serum TB level in the patients who underwent EBD and received CS was significantly higher than that in patients treated with CS alone ($p < 0.001$). There was no significant difference in age, sex, serum IgG or IgG4 level, or the presence of extrapancreatic biliary stricture between the two treatment groups. Forty-one patients treated with CS alone did not undergo EBD for the following reasons: no or mild elevation of biliary enzyme levels without obstructive jaundice in 31 patients; physician's judgement that CS initiation should be preferred over EBD despite obstructive jaundice with a serum TB level of $> 3.0$ mg/dL in 9 patients; and EBD failure because of obstructive jaundice due to type 2 diffuse intrahepatic biliary stricture in 1 patient (Fig 1). The 10 patients with obstructive jaundice had no symptoms of suppurative cholangitis at the time of CS initiation and achieved clinical improvement without biliary tract infection.

## Clinical relapse and biliary tract complications

During the follow-up, relapse of ISC occurred in 11 patients (15.9%) after CS-induced improvement. Table 4 shows the clinical characteristics of the patients who did and did not relapse. No clinical factor was significantly predictive of ISC relapse before CS initiation. Neither maintenance therapy ($p = 0.797$) nor a daily maintenance dose of prednisolone ($p = 0.320$) was associated with relapse. However, patients who discontinued CS during the clinical course were more likely to experience relapse than those who continued CS maintenance therapy ($p = 0.089$).

Regarding biliary tract complications during the follow-up, three patients (4.3%), all of whom had undergone EBD, developed bile-duct stones, while none of those treated with CS alone developed bile-duct stones ($p = 0.032$) (Table 3). Furthermore, 2 of the 3 patients had not undergone stent removal, compared to 1 of 25 patients without bile-duct stones ($p < 0.001$) (Table 5). The median duration of EBD in patients with bile-duct stones was 7.9

**Table 3. The differences in clinical characteristics between the patients treated with CS alone and CS in addition to EBD.**

| Parameter ($n = 69$) | CS alone ($n = 41$) | EBD and CS ($n = 28$) | $p$ value |
|---|---|---|---|
| Age, mean ± SD, years | 65.2 ± 10.1 | 67.4 ± 11.2 | 0.285 |
| Sex, male / female | 36 / 5 | 24 / 4 | 0.800 |
| Obstructive jaundice[†], present / absent | 10 / 31 | 22 / 6 | $< 0.001$ |
| Serum IgG level, median (range), mg/dl | 1826 (950–5963) | 1945 (1191–2968) | 0.707 |
| Serum IgG4 level, median (range), mg/dl | 465.5 (145–4080) | 547 (148–3805) | 0.348 |
| Extrapancreatic biliary strictures, present / absent | 9 / 32 | 5 / 23 | 0.678 |
| Development of bile-duct stones, present / absent | 0 / 41 | 3 / 25 | 0.032 |
| Development of biliary tract cancer, present / absent | 0 / 41 | 0 / 28 | N/A |

† Serum total bilirubin level of $> 3.0$ mg/dL with dilation of bile-duct.

EBD, endoscopic biliary drainage; CS, corticosteroid therapy, N/A; not available.

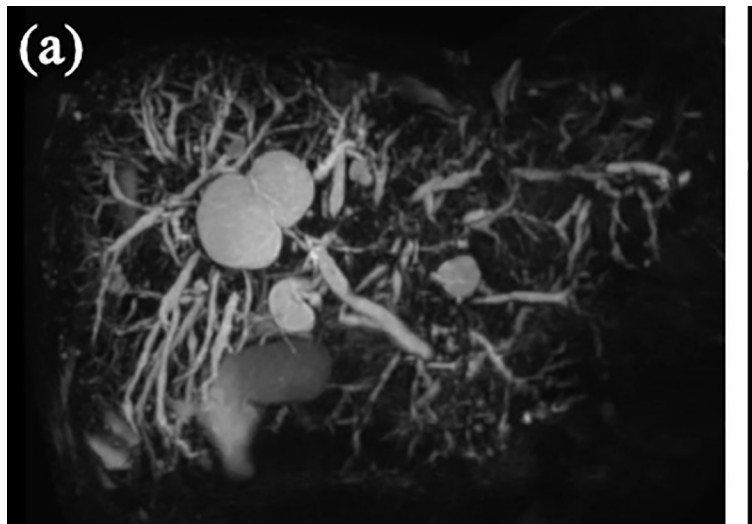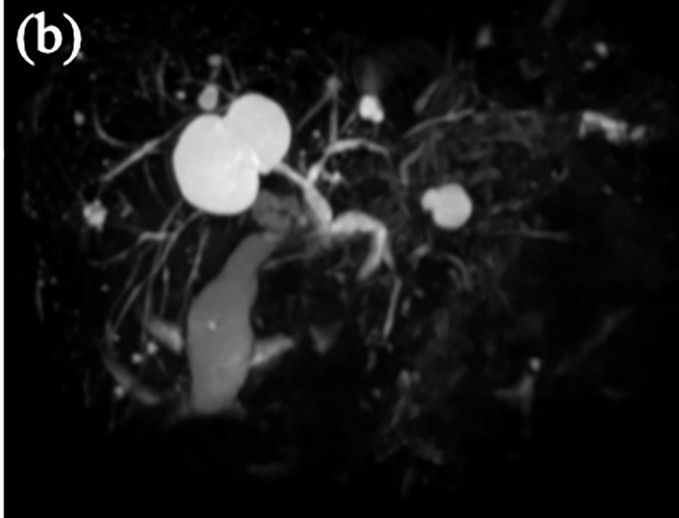

**Fig 1. Magnetic resonance cholangiopancreatography images of a patient with IgG4-related sclerosing cholangitis who showed obstructive jaundice.** (a) An image obtained before corticosteroid (CS) initiation showing both intrapancreatic and diffuse intrahepatic biliary strictures with a serum total bilirubin (TB) level of 11.1 mg/dL. Accordingly, endoscopic biliary drainage was considered ineffective. (b) An image obtained 2 months after CS initiation showing cholangiopancreatographic improvement. The serum TB level decreased to 2.1 mg/dL, without symptoms of biliary tract infection.

months, which is longer than that in patients without bile duct-stones (1.5 months, $p = 0.063$). Liver abscess occurred in one patient who had not undergone stent removal after CS-induced improvement. None of the patients developed biliary tract cancer during the follow-up.

## Discussion

In large-scale retrospective studies, $> 70\%$ of cases of obstructive jaundice secondary to AIP received EBD before CS initiation to prevent biliary tract infection[6,7]. Although some reports and reviews have described the short- or long-term prognosis of biliary stricture

**Table 4. The differences in clinical characteristics between the patients who experienced relapse and not.**

| Parameter ($n$ = 69) | Relapse (+) ($n$ = 11) | Relapse (-) ($n$ = 58) | $p$ value |
|---|---|---|---|
| Age, mean ± SD, years | 71.1 ± 11.5 | 65.1 ± 10.1 | 0.103 |
| Sex, male / female | 9 / 2 | 51 / 7 | 0.581 |
| Obstructive jaundice[†], present / absent | 5 / 6 | 27 / 31 | 0.947 |
| Serum IgG level, median (range), mg/dl | 1772 (950–3010) | 1910 (1191–5963) | 0.140 |
| Serum IgG4 level, median (range), mg/dl | 423 (145–1740) | 538 (155–4080) | 0.157 |
| Extrapancreatic biliary strictures, present / absent | 3 / 8 | 11 / 47 | 0.530 |
| Treatment with EBD, present / absent | 5 / 6 | 23 / 35 | 0.720 |
| Initial dose of prednisolone, median (range), mg/day | 30 (20–40) | 30 (5–45) | 0.288 |
| CS maintenance therapy[‡], present / absent | 10 / 1 | 54 / 4 | 0.797 |
| Maintenance dose of prednisolone, median (range), mg/day | 4 (1–10) | 5 (1–10) | 0.320 |
| Maintenance prednisolone dose of $\geq$ 5 mg/day, present / absent | 5 / 6 | 40 / 18 | 0.133 |
| CS discontinuation, present / absent | 7 / 4 | 21 / 37 | 0.089 |

† Serum total bilirubin level of $> 3.0$ mg/dL with dilation of bile-duct.

‡ Maintenance therapy was defined as continuous administration of CS for $> 6$ months after initiation.

EBD, endoscopic biliary drainage; CS, corticosteroid therapy.

**Table 5. The differences in the management of EBD between the patients who developed bile-duct stones and not.**

| Parameter ($n$ = 28) | Bile-duct stones (+) ($n$ = 3) | Bile-duct stones (-) ($n$ = 25) | $p$ value |
|---|---|---|---|
| Prolonged biliary stenting, present / absent | 2 / 1 | 1 / 24 | <0.001 |
| The duration of EBD, median (range), months | 7.9 (1.7–10.1) | 1.5 (0.3–31.6) | 0.063 |

EBD, endoscopic biliary drainage.

accompanying AIP, few reports focused on ISC irrespective of the presence of AIP. Japanese researchers recently published clinical practice guidelines for ISC[9], which describe the appropriate management of biliary stricture in patients with ISC. However, some aspects of ISC treatment by EBD followed by CS are unclear.

The clinical and radiological features of ISC are usually resolved by CS, which is typically followed by safe removal of the stent. If the stent is not removed, improvement of the biliary stricture due to amelioration of inflammation of the bile-duct wall could cause spontaneous stent dislodgement, which is associated with a risk of intestinal perforation[10]. Therefore, stent removal after CS-induced improvement, and the timing thereof, is important. In this study, stent dislodgement occurred in 42.9% of the patients. Notably, in most of those patients stent dislodgement was estimated to occur 2 weeks to 2 months after the CS initiation. On the other hand, premature stent removal may cause recurrence of obstructive jaundice due to residual biliary stricture if the CS-induced improvement is insufficient. It is preferable to avoid repeat EBD after stent removal from the point of view of procedure-related costs and the risk of complications. In this study, none of patients who underwent intentional stent removal after CS-induced improvement of biliary stricture caused by ISC required repeat EBD due to early recurrence of obstructive jaundice within 2 weeks. Therefore, we suggest that stent removal should be performed within 2 weeks after CS initiation in patients with obstructive jaundice. If CS-induced improvement of biliary stricture seemed to be insufficient on ERC after stent removal, stent reinsertion might be necessary to prevent recurrence of obstructive jaundice. In that case, it is important to determine the appropriate timing for stent removal with closely monitoring. Noteworthy, two patients developed spontaneous stent dislodgement before CS initiation in our study. Although we placed a straight-type plastic stent in patients with ISC, the insertion of a double-pigtail plastic stent would likely be safer because of the lower risk of stent dislodgement and related complications. ENBD is also useful and safe for treating purulent cholangitis and/or multiple biliary strictures like type 2–3, while endoscopic ultrasound guided-fine needle aspiration (EUS-FNA), which is necessary for pathological differentiation from pancreatic cancer in patients with focal-type AIP, is difficult to perform during insertion of ENBD. Each method of EBD has advantages and disadvantages as mentioned above, so an appropriate strategy of EBD should be selected according to the patient's condition.

Ten patients with obstructive jaundice achieved improvement with CS alone in this study, and none of them developed purulent cholangitis. Clinical practice guidelines for ISC recommend performing EBD in cases involving obstructive jaundice, while they also state that CS can be initiated without EBD in cases involving mild jaundice without purulent cholangitis in patients for whom the diagnosis of ISC is definite and pathological examinations for biliary stricture are unnecessary[9,11,12]. Some reports indicated obstructive jaundice secondary to AIP may be safely managed using CS alone[13,14]. In our opinion, prior CS initiation may be acceptable in patients with obstructive jaundice caused by multiple biliary stricture like type 2–3 for which effective EBD is impossible as shown in Fig 1, of course, under the condition of close monitoring by an experienced physician. However, there are a small number of studies

about treatment with CS alone for ISC involving obstructive jaundice, and we cannot conclude the safety of it. Furthermore, patients with ISC who undergo ERC followed by transpapillary bile-duct biopsy and/or brushing cytology should be treated with short term EBD to relieve cholestasis and prevent cholangitis even without obstructive jaundice. Of course, in patients who did not undergo ERC at the time of initial diagnosis, if radiological or laboratory findings do not improve rapidly even after CS initiation, ERC should be performed again to determine the validity of the pretreatment diagnosis[13,15].

We also investigated the long-term prognosis, including the incidence of clinical relapse and biliary tract complications, of patients with ISC. Reports from Western countries suggest that relapse occurs in 30–57% of patients during CS maintenance therapy or after CS discontinuation[7,16]. In a Japanese retrospective cohort study, relapse of biliary stricture occurred in 19% of patients[4], similar to our finding. The difference in results between Western and Japanese studies seems to be due to the duration of CS administration. Patients received CS for a median of 5.5 months, and only 47% received CS for > 6 months, in a large cohort study from the United Kingdom[16]. The consensus in Japan is that long-term (3-year) prednisolone maintenance therapy at a daily dose of approximately 5 mg is desirable to prevent relapse[17]. Furthermore, Hirano *et al*. recommended that long-term maintenance therapy with CS should be continued beyond 3 years, because relapse occurred in 48% of patients with AIP who discontinued CS after 3 years in a prospective trial[18]. In this study, patients who discontinued CS maintenance therapy were more likely to experience relapse.

Prediction of ISC relapse based on the clinical characteristics of patients before CS initiation would enable the appropriate duration of CS administration to be determined, with the aim of preventing relapse. Although no clinical factor has been shown to predict ISC relapse, some risk factors have been suggested, including a high serum IgG4 level, the presence of extrapancreatic or multiple biliary strictures, and a thickened bile-duct wall during the initial attack [7,19,20]. Unfortunately, in this study no clinical factor was significantly predictive of relapse before CS initiation. Considering the difficulty of identifying patients who will experience ISC relapse based on a single clinical value, a system for scoring the disease activity of ISC based on multiple parameters with high sensitivity and specificity is needed for predicting relapse. This would enable identification of patients at high risk of relapse, to whom long-term CS maintenance therapy should be administered.

Biliary tract complications may become a clinical issue during the follow-up. Our results revealed that bile-duct stones were likely to develop in patients treated with EBD, particularly if biliary stenting was prolonged. ISC causes chronic inflammation of the bile-duct wall and its accessory glands, while the injury to the bile-duct epithelium itself is relatively mild[3,21]. The mucosal surface of the bile-duct in patients with ISC is macroscopically smooth (and its lumen patent), unlike in those with primary sclerosing cholangitis or cholangiocarcinoma. Therefore, the risk of bile-duct stones in patients with ISC is relatively low after improvement of cholestasis by CS. In this study, patients treated with CS alone did not develop bile-duct stones. Erosions or ulcers in the bile-duct epithelium of patients with ISC frequently result from long-term biliary stenting and increase the risk of bile-duct stones due to sludge formation. In fact, the biliary stent remained *in situ* for a prolonged period in patients who developed bile-duct stones in this study. However, the long-term incidence of bile-duct stones in patients with ISC is unclear. Since three patients with bile-duct stones are too few for statistical analysis, the result may vary if patients with ISC increase and are observed for a longer follow-up period. One patient who had not undergone stent removal developed liver abscess during CS maintenance therapy. Both EBD-related retrograde cholangitis and CS-induced immunosuppression may lead to liver abscess. The long-term management of ISC requires caution because biliary tract infection is likely to develop in patients who undergo EBD. In this study, none of the

patients developed biliary tract cancer during the follow-up. While there have been previous reports of cholangiocarcinoma concurrent with ISC at disease onset, there has been no case report of the development of biliary tract cancer during the course of ISC[22–25]. Kamisawa *et al.* reported the presence of KRAS mutations in epithelial cells in the pancreas, common bile-duct, and gallbladder of patients with AIP[26]. Although ISC complicated by bile-duct cancer is rare, clinicians should be aware of the possibility.

This study had several limitations. First, it was not a prospective study. Second, the results were derived from a small number of patients seen at only a few institutions, which could have caused bias. Third, there is no standardized indication for EBD, initial and maintenance doses of prednisolone. Especially with regard to EBD, we enrolled only patients treated with EBD by placement of plastic stent for before CS initiation. However, ENBD may be suitable for patients with symptoms of purulent cholangitis. Furthermore, there may be differences in the optimal strategy for EBD between type 1 ISC and the others including type 2, 3 and 4, which have extrapancreatic biliary stricture. Showing the results of this study using this retrospective cohort may not be convincing. To find the best treatment method, this study should have used a prospective observational design.

In conclusion, we investigated recent advances in the treatment of ISC. Biliary stent removal should be carried out within 2 weeks after CS initiation if CS improves biliary stricture to prevent spontaneous stent dislodgement. Obstructive jaundice due to biliary stricture can be treated safely by CS alone in limited patients without purulent cholangitis. Long-term management requires caution because bile-duct complications are likely in patients who undergo EBD. Therefore, further prospective investigations are needed to evaluate the disease activity of ISC and methods for reducing the risk of biliary tract complications.

## Author Contributions

**Data curation:** Masaki Miyazawa.

**Formal analysis:** Masaki Miyazawa.

**Investigation:** Masaki Miyazawa, Hajime Takatori, Kazunori Kawaguchi, Kazuya Kitamura, Kuniaki Arai, Koichiro Matsuda, Takeshi Urabe, Katsuhisa Inamura, Takuya Komura, Hideki Mizuno, Uichiro Fuchizaki, Taro Yamashita, Yoshio Sakai, Tatsuya Yamashita, Eishiro Mizukoshi, Masao Honda, Shuichi Kaneko.

**Methodology:** Masaki Miyazawa.

**Project administration:** Masaki Miyazawa.

**Supervision:** Hajime Takatori, Kazunori Kawaguchi, Kazuya Kitamura, Kuniaki Arai, Taro Yamashita, Yoshio Sakai, Tatsuya Yamashita, Eishiro Mizukoshi, Masao Honda, Shuichi Kaneko.

**Validation:** Masaki Miyazawa.

**Visualization:** Masaki Miyazawa.

**Writing – original draft:** Masaki Miyazawa.

**Writing – review & editing:** Masaki Miyazawa.

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
