## [Decision Letter · Decision Letter 0]

12 Mar 2020

PONE-D-20-06103

Management of biliary stricture in patients with IgG4-related sclerosing cholangitis

PLOS ONE

Dear Dr. Masaki Miyazawa,

Thank you for submitting your manuscript to PLOS ONE. After careful consideration, we feel that it has merit but does not fully meet PLOS ONE’s publication criteria as it currently stands. Therefore, we invite you to submit a revised version of the manuscript that addresses the points raised during the review process.

We would appreciate receiving your revised manuscript within 60 days. To enhance the reproducibility of your results, we recommend that if applicable you deposit your laboratory protocols in protocols.io, where a protocol can be assigned its own identifier (DOI) such that it can be cited independently in the future. For instructions see: http://journals.plos.org/plosone/s/submission-guidelines#loc-laboratory-protocols

We look forward to receiving your revised manuscript.

Kind regards,

Gianfranco D. Alpini

Academic Editor

PLOS ONE

Journal Requirements:

2)  We note that you have indicated that data from this study are available upon request. PLOS only allows data to be available upon request if there are legal or ethical restrictions on sharing data publicly. For information on unacceptable data access restrictions, please see http://journals.plos.org/plosone/s/data-availability#loc-unacceptable-data-access-restrictions.

Reviewers' comments:

Reviewer's Responses to Questions

**Comments to the Author**

1. Is the manuscript technically sound, and do the data support the conclusions?

Reviewer #1: Yes

Reviewer #2: Yes

2. Has the statistical analysis been performed appropriately and rigorously? 

Reviewer #1: Yes

Reviewer #2: No

3. Have the authors made all data underlying the findings in their manuscript fully available?

Reviewer #1: Yes

Reviewer #2: Yes

4. Is the manuscript presented in an intelligible fashion and written in standard English?

Reviewer #1: Yes

Reviewer #2: Yes

5. Review Comments to the Author

Reviewer #1: This retrospective study is neat with great clinical significance, regarding to the management of biliary stricture in IgG4-related sclerosing cholangitis (ISC). It presents a profile of 69 patients with ISC, who underwent corticosteroid (CS) therapy or endoscopic biliary drainage (EBD) combined with CS. ISC is a rare disease that belongs to a subgroup of sclerosing cholangitis. The diagnosis of ISC is complicate and difficult. Distinguish from PSC, cholangiocarcinoma and pancreatic cancer should always be considered. The treatment is not easy even though CS is commonly effective. In ISC patients with biliary obstruction and jaundice, EBD may play an important role except for CS. This research focused on this special situation and indicated the safety and efficacy of two treatment methods.

Due to the low prevalence of ISC, it is difficult to design and undertake prospective randomized clinical trials. A well-designed retrospective study is valuable for the clinical reference. The sample size of this study is large enough considering the rarity of ISC. The criteria for the enrollment and exclusion are rigorous. The statistics analysis is sound. The results are convincing and described in every detail. In the discussion section, it shows the novelty of this study and the differences from some others. Overall, this article would be useful for specialists carrying out EBD combined with CS therapy.

I have only two questions/suggestions for the manuscript:

1. The abstract was written not clearly enough, sometimes a little confusing. I don’t understand it until I read through the paper, especially the following descriptions in the results. The manuscript would be perfect if the abstract is modified properly.

2. The authorship is confusing. Did not see the “*” after author’s names. All “¶” contributed equally? First 5 authors and last 6 authors?

Reviewer #2: In this study, the authors aimed to determine the optimal approach with endoscopic biliary drainage (EBD) and corticosteroid (CS) for the treatment of lgG4-related sclerosing cholangitis (ISC). 69 patients with ISC treated with CS were enrolled. Twenty-eight patients (40.6%) were treated with EBD for biliary stricture before CS initiation. Stent removal was carried out safely without early (within 2 weeks) recurrence of obstructive jaundice in all eleven patients who underwent stent removal within a month of CS initiation. Ten patients (35.7%) experienced spontaneous stent dislodgement after CS initiation, and seven (70%) of them developed stent dislodgement 2 weeks to 2 months after CS initiation. All ten patients with obstructive jaundice treated with CS alone did not develop biliary tract infection. All three patients with bile duct stones had received EBD. Long-term biliary stenting was a risk factor for bile duct stones. The manuscript is very well written, and the data clearly support the conclusions.

1. My concern is that sample size is too small, only three patients who had undergone EBD, developed bile-duct stones which is hard to draw rigorous conclusions.

2. Logistic regression can predict the risk of developing a given disease, which is based on observed characteristics of the patient. Therefore, the authors should use Logistic regression to predict the risk for bile duct stones.

6. PLOS authors have the option to publish the peer review history of their article (what does this mean?). If published, this will include your full peer review and any attached files.

Reviewer #1: No

Reviewer #2: No

---

## [Author Response · Author response to Decision Letter 0]

1 Apr 2020

Response to Reviewers

Reviewer #1:

1. The abstract was written not clearly enough, sometimes a little confusing. I don’t understand it until I read through the paper, especially the following descriptions in the results. The manuscript would be perfect if the abstract is modified properly.

• I modified the descriptions in the results to make the abstract easier to understand.

2. The authorship is confusing. Did not see the “*” after author’s names. All “¶” contributed equally? First 5 authors and last 6 authors?

• I rewrote the title page as shown in the policy. Symbol * means corresponding author and was marked in author’s name. Roles of authors with symbol ¶ were investigation and supervision. A Role of authors with symbol & was investigation. 

Reviewer #2:

1. My concern is that sample size is too small, only three patients who had undergone EBD, developed bile-duct stones which is hard to draw rigorous conclusions.

• As you say, I think the sample size is too small for three patients who developed bile-duct stones. The result may vary if patients of ISC increase and are observed for longer follow-up period. I add the above statement to the discussion part.

2. Logistic regression can predict the risk of developing a given disease, which is based on observed characteristics of the patient. Therefore, the authors should use Logistic regression to predict the risk for bile duct stones.

• The results of univariate analysis to determine the risk factors for bile-duct stones showed that EBD administration, sustained biliary stenting and duration of EBD were risk factors. Other clinical parameters were not risk factors of bile-duct stones. Logistic regression analysis was attempted, but the number of patients who developed bile-duct stones was too small to be analyzed. The result may vary if patients of ISC increase.

---

## [Decision Letter · Decision Letter 1]

8 Apr 2020

Management of biliary stricture in patients with IgG4-related sclerosing cholangitis

PONE-D-20-06103R1

Dear Dr. Masaki Miyazawa,

We are pleased to inform you that your manuscript has been judged scientifically suitable for publication and will be formally accepted for publication once it complies with all outstanding technical requirements.

With kind regards,

Gianfranco D. Alpini

Academic Editor

PLOS ONE

Additional Editor Comments (optional):

Reviewers' comments:

Reviewer's Responses to Questions

**Comments to the Author**

1. If the authors have adequately addressed your comments raised in a previous round of review and you feel that this manuscript is now acceptable for publication, you may indicate that here to bypass the “Comments to the Author” section, enter your conflict of interest statement in the “Confidential to Editor” section, and submit your "Accept" recommendation.

Reviewer #1: All comments have been addressed

2. Is the manuscript technically sound, and do the data support the conclusions?

Reviewer #1: Yes

3. Has the statistical analysis been performed appropriately and rigorously? 

Reviewer #1: Yes

4. Have the authors made all data underlying the findings in their manuscript fully available?

Reviewer #1: Yes

5. Is the manuscript presented in an intelligible fashion and written in standard English?

Reviewer #1: Yes

6. Review Comments to the Author

Reviewer #1: (No Response)

7. PLOS authors have the option to publish the peer review history of their article (what does this mean?). If published, this will include your full peer review and any attached files.

Reviewer #1: No

---

## [Editor Report · Acceptance letter]

21 Apr 2020

PONE-D-20-06103R1 

Management of biliary stricture in patients with IgG4-related sclerosing cholangitis 

Dear Dr. Miyazawa:

I am pleased to inform you that your manuscript has been deemed suitable for publication in PLOS ONE. Congratulations! Your manuscript is now with our production department. 

With kind regards,

on behalf of

Dr. Gianfranco D. Alpini 

Academic Editor

PLOS ONE